# Improved Transformer-Based Deblurring of Commodity Videos in Dynamic Visual Cabinets

**Shuangyi Huang** [1,†]**, Qianjie Liang** [1,†]**, Kai Xie** [1,*]**, Zhengfang He** [2]**, Chang Wen** [3]**, Jianbiao He** [4] **and Wei Zhang** [5]

1  School of Electronic Information and Electrical Engineering, Yangtze University, Jingzhou 434023, China; shuangyi_huang.stu@yangtzeu.edu.cn (S.H.); liangqianjie.stu@yangtzeu.edu.cn (Q.L.)
2  Zhejiang Uniview Technologies Co., Ltd., Hangzhou 310051, China; hezhengfang@uniview.com
3  School of Computer Science, Yangtze University, Jingzhou 434023, China; 400100@yangtzeu.edu.cn
4  School of Computer Science, Central South University, Changsha 410083, China; jbhe@mail.csu.edu.cn
5  School of Electronic Information, Central South University, Changsha 410083, China; csuzwzbn@csu.edu.cn
*  Correspondence: xiekai@yangtzeu.edu.cn; Tel.: +86-136-9731-5482
†  These authors contributed equally to this work.

**Abstract:** In the dynamic visual cabinet, the occurrence of motion blur when consumers take out commodities will reduce the accuracy of commodity detection. Recently, although Transformer-based video deblurring networks have achieved results compared to Convolutional Neural Networks in some blurring scenarios, they are still challenging for the non-uniform blurring problem that occurs when consumers pick up the commodities, such as the problem of difficult alignment of blurred video frames of small commodities and the problem of underutilizing the effective information between the video frames of commodities. Therefore, an improved Transformer video deblurring network is proposed. Firstly, a multi-scale Transformer feature extraction method is utilized for non-uniform blurring. Secondly, for the problem of difficult alignment of small-item-blurred video frames, a temporal interactive attention mechanism is designed for video frame alignment. Finally, a feature recurrent fusion mechanism is introduced to supplement the effective information of commodity features. The experimental results show that the proposed method has practical significance in improving the accuracy of commodity detection. Moreover, compared with the recent Transformer deblurring algorithm Video Restoration Transformer, the Peak Signal-to-Noise Ratio of this paper's algorithm is higher than that of the Deep Video Deblurring dataset and the Fuzzy Commodity Dataset by 0.23 dB and 0.81 dB, respectively.

**Keywords:** motion blur; video deblurring; multi-scale Transformer; temporal interaction attention mechanism; feature recurrent fusion mechanism

## 1. Introduction

With the rapid development of artificial intelligence technology and social progress, unmanned retail has made rapid development. From traditional spring machines, crawler lifts, Radio Frequency Identification unmanned vending machines, gravity cabinets, and static cabinets, to the recent hot dynamic cabinets, the field of unmanned retail is constantly innovating [1]. A dynamic vision cabinet is based on the current deep learning technology in the field of artificial intelligence, such as the birth of the unmanned retail vending cabinet. It is equipped with an advanced computer vision technology (target detection, image recognition, target tracking, and other application algorithms) camera to monitor the picking up of the cabinet of goods by a consumer in real time. After obtaining the video of the process of consumers picking up goods, the camera will track the movement trajectory of the goods held by the consumer, detect the shape, color, and other characteristics of the goods until the end of consumption, and determine the category and quantity of the goods purchased by the consumer so as to carry out automatic deduction of expenses. A dynamic

vision cabinet is used by the consumer to take the goods for detection, and this process is very mature, but there are still some problems to be solved.

When performing merchandise detection in vision cabinets [2,3], it is usually necessary to feed the merchandise picked up by the consumer into the target detection network for detection. However, in this process, the consumer's rapid picking up of the merchandise can lead to motion blurring of the merchandise under fixed background conditions. When the detection network detects these blurred merchandise features, it may reduce the detection accuracy or even fail to detect the merchandise. Therefore, the elimination of blurring generated by moving commodities is a technical problem to be solved for commodity detection.

Traditional image deblurring utilizes methods based on fuzzy kernel estimation [4], while as deep learning has advanced, numerous techniques based on Convolutional Neural Networks (CNNs) [5] have been put out to address the issue. However, video deblurring [6,7] necessitates efficient spatio-temporal modeling to extract the complementary information found in the adjacent frames to aid in better recovery, in contrast to single-picture deblurring [8].

In video deblurring methods, Su et al. [9] introduced a network of encoders and decoders that stacks multiple consecutive frames as inputs and directly outputs the recovered potential frames to learn how to accumulate information and deblur the video. Kim et al. [10] developed a deep recurrent network that periodically recovers the potential frames by connecting multiple frame features. For better temporal information, Zhang et al. [11] developed spatio-temporal 3D convolution to help potential frame recovery, and Pan et al. [12] introduced a temporal sharpness to enhance the deblurring network's performance. Although these methods can continuously improve the deblurring performance, some problems cannot be solved well in the task of this paper. For example, when a consumer is picking up a commodity, the different picking up speeds result in different degrees of movement of the commodity, which can cause non-uniform blurring of the commodity to occur.

In video deblurring, video frame alignment is essential in order to utilize the effective information between adjacent frames. In recent years, regarding the task of video frame alignment, Zhou et al. [13] used dynamic filters to align consecutive frames, Wang et al. [14] achieved better alignment performance based on deformable convolution, and Li et al. [15] utilized self-attention to capture pixel correlation of consecutive frames. However, CNN-based spatio-temporal modeling capabilities are limited, and the alignment task remains challenging.

Following its impressive remote and relational modeling demonstrations in Natural Language Processing in recent years, a Transformer [16]—also known as a Vision Transformer—has been progressively integrated into Computer Vision [17–19]. One of the recent breakthroughs in Vision Transformers is the Pyramid Vision Transformer [20] and the hierarchical design of the Swin Transformer [21].

Beyond the current CNN-based backbones, these Transformer-based backbones have prompted researchers to investigate the use of Transformers in the restoration industry. Liang and associates [22] proposed a Temporal Attention Mechanism (TMSA) to align neighboring frames, and they exploited the excellent spatio-temporal modeling capability of the Transformer to extract spatio-temporal features that achieved good results in dealing with blurring.

However, in the visual cabinet task in this paper, all the above methods make it difficult to align the merchandise video frames when blurring occurs due to the low resolution of the small merchandise. Moreover, these methods can only establish a connection between neighboring frames when performing the task of aligning commodity video frames and cannot model with more distant video frames, which will lead to partial loss of commodity information.

To address the above problems and remove the fuzzy features of commodities in the visual cabinet effectively, this paper proposes a new Transformer-based video deblurring network, which includes the following steps:

1.  A multi-scale Transformer structure is defined to deal with the non-uniform blur caused by different degrees of motion when picking up the goods;
2.  A temporal interactive attention mechanism (TI-MSA) is proposed based on the remote modeling relation of the Transformer to solve the problem that some blurred video frames of small commodities cannot be aligned;
3.  A feature recurrent fusion mechanism is proposed to construct the global modeling relation of commodity video frames to solve the problem of losing valid information in reconstructed commodity video frames.

This paper is organized as follows. The general algorithm and associated ideas are presented in Section 2. The experimental platform, experimental dataset, comparison experiments, ablation experiments, visualization, and analysis of the experimental results are all covered in detail in Section 3. In Section 4, the algorithm's summary is outlined.

## 2. The Algorithms

A flowchart of the algorithm is shown in Figure 1. It is divided into three parts, namely, video frame feature extraction, video frame alignment and fusion, and video frame reconstruction module. (1) Take the input video frame and go through the multi-scale Transformer module to perform feature extraction; (2) feed the feature-extracted video frame into the temporal interactive attention mechanism and the feature recurrent fusion mechanism to perform alignment and fusion; and (3) reconstruct the features obtained by the temporal interactive attention fusion to reconstruct the obtained features into clear video frames.

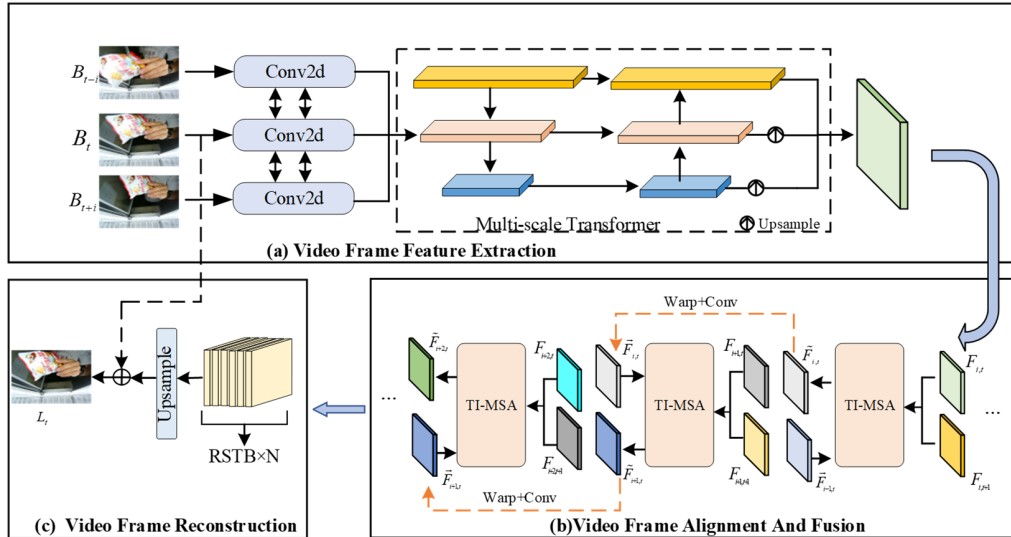

**Figure 1.** Overall flow chart of the algorithm. (**a**) The blurred merchandise video frames will be extracted with shallow features by a convolutional layer and then fed into the multi-scale Transformer structure for deep feature extraction. (**b**) After obtaining the feature-extracted video frames, the temporal cross-attention module will fuse the information of neighboring video frames, and the fused video frame features will be passed to the next layer of the temporal cross-attention structure after the convolution and warp operations are passed to the next layer of the temporal cross-attention module. In this way, a feature cycle fusion is constructed to supplement the effective features. (**c**) The aggregated features from multiple frames are fed into the Residual Swin Transformer Block (RSTB) for the upsampling operation and are then fused with the original features using the global residual learning strategy and, finally, reconstructed into clear video frames. The purpose of the upsampling operation in the figure is to zoom in on the feature map.

### 2.1. Video Frame Feature Extraction

#### 2.1.1. RSTB

Because of the great success of SwinIR [23] in image reconstruction, this paper adopts RSTB to facilitate a more effective content-based interaction between attention weights and image content.

In Figure 2, the features $F \in R^{W \times H \times d}$ are set as inputs to the Swin Transformer Block, which is divided into non-overlapping windows of spatial size m × m by a reshaping operation. After that, a multi-head self-attention is executed in each window. For a feature f that performs a multi-head self-attention in a window, a learnable positional encoding $X \in R^{m \times m \times d}$ is first added to it, and then the transformation matrix $W_Q$, $W_K$, and $W_V$ is utilized to generate a query, key, and value via linear projection.

$$Q = XW_Q, K = XW_K, V = XW_V \tag{1}$$

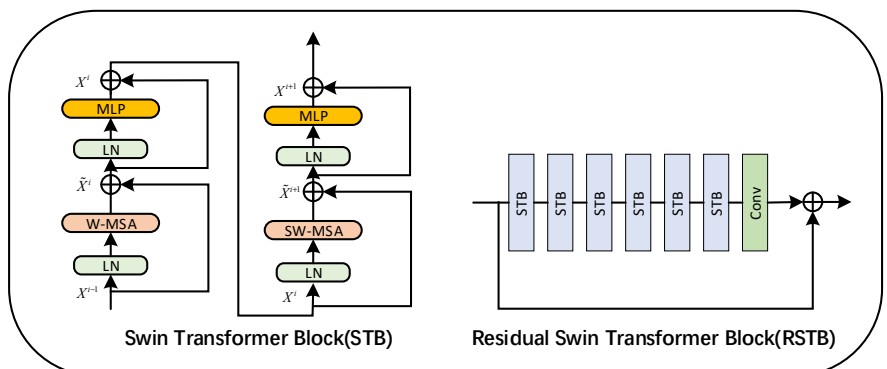

**Figure 2.** Schematic diagram of the RSTB.

In general, $Q, K, V \in R^{m \times m \times d}$, and $m$ is the number of patches in the window and $d$ is the dimension. The attention matrix computed by the self-attention mechanism within a localized window is

$$Attention(Q, K, V) = SoftMax\left(\frac{QK^T}{\sqrt{d}}\right)V \tag{2}$$

The original Vision Transformer performs multi-head attention at all locations in space with a computational complexity proportional to the quadratic counterpart of the image size ($O = 2(HW)^2 d$). Due to its high computational complexity, it is difficult to handle high-resolution images, especially in video deblurring tasks. However, in the multi-head self-attention mechanism of the Swin Transformer Block, its computational complexity is $O(W - MSA) = 2m^2 HWd$, and its computational amount increases linearly with the size of the image ($W \times H$), which is much smaller compared to the computational complexity of Vision Transformer's multi-head self-attention that increases quadratically with the image size. In this way, the Transformer can greatly reduce the computational cost of maintaining a high-resolution representation without severely degrading the sampling of the image or feature map. Secondly, the operation of the Swin Transformer and its shift window can enhance the Transformer's remote-dependent modeling capability for potential frame reconstruction.

In the RSTB shown in Figure 2, each RSTB utilizes multiple Swin Transformer layers for localized attention mechanisms and inter-window interactions. Moreover, a convolutional layer is added at the end of the block to enhance the features and provide a convenient path for feature aggregation through residual concatenation.

For the input features $F_{i,0}$ of the *i*-th RSTB, this paper will extract the intermediate features passing through the Swin Transformer Block as $F_{i,1}, F_{i,2}, \cdots, F_{i,j}$. The formula is as follows:

$$F_{i,j} = H_{STB_{i,j}}(F_{i,j-1}), j = 1, 2, \ldots, L, \tag{3}$$

$H_{STB_{i,j}}(\cdot)$ in Equation is the *j*-th Swin Transformer Block in the *i*-th RSTB. Then, the convolutional layer is added before the remaining connections. The output of the RSTB is given by the following Equation:

$$F_{i,out} = H_{CONV_i}(F_{i,L}) + F_{i,0} \tag{4}$$

where $H_{CONV_i}(\cdot)$ is the convolutional layer in the *i*-th RSTB; this design allows for different levels of feature aggregation and handles the variation of video frames in terms of panning well.

### 2.1.2. Multi-Scale Transformer

In the feature extraction stage, first, in order to better utilize the feature information present in the adjacent frames, in this paper, $2T + 1$ consecutive video frames $B_{[t-T,t+T]}$ (*T* is the number of future frames and past frames at the current moment *t*) are used as inputs, where the intermediate frame $B_t$ is used as the reference frame. After that, a $3 \times 3$ convolution is used for shallow feature extraction. This allows access to the low-frequency information of the video frames, and the convolutional layer performs well in early visual processing, leading to more stable optimization and better results. Next, in order to refine the features, each $B_{t+i} \in R^{W \times H \times C}$ (*W*, *H*, and *C* are the width, height, and number of channels of the input frame, respectively) blur frame is divided into non-overlapping patches by referring to the patch Embedding operation in VIT [18]. Here, a $4 \times 4$ patch size is used for the segmentation, which generates finer-grained features. Finally, it is projected into a vector of dimension *d* (*d* = 256 in this paper's algorithm) by a linear transformation.

In Figure 3, the multi-scale Transformer proposed in this paper mainly consists of an encoder and decoder structure consisting of RSTBs, which are utilized to process motion blur features of commodities with different degrees of motion. For the encoding branch, the features undergo a Patch Reduction operation after passing through the RSTB, and the encoder has a total of two such operations. For each Patch Reduction, the number of channels is increased to twice the original number, spatial downsampling is performed, and finally, a linear transformation is used to project the feature dimensions to the d-dimension.

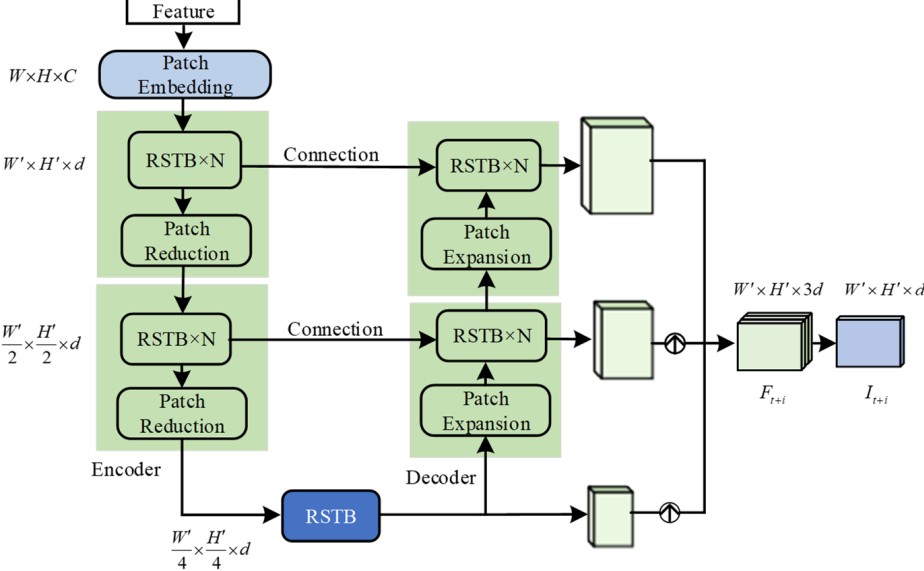

**Figure 3.** Multi-scale Transformer.

When the features pass through the encoder, at this time the network does not converge well because the Transformer is too deep. Therefore, before entering the decoder, this paper adopts an RSTB to construct a bottleneck to learn deep features; at this time, the dimension and resolution of the features remain unchanged.

In the decoding branch, the features after the bottleneck structure will enter the RSTB through the Patch Expansion operation and then fuse with the features from the corresponding encoder layer to further refine the features. In this stage, two Patch Expansion operations are performed. Unlike the Patch Reduction operation, for each Patch Expansion, the number of channels of the feature is doubled, and then the small-scale features are spatially upsampled and linearly transformed to project to the d-dimension. Throughout the module, the number of RSTBs passing through each upsampling and downsampling is $N = 3$.

In the encoder–decoder operation, the expression is as follows:

$$B'_{t+i} = R_{pr}\left(B^n_{t+i}\right) \tag{5}$$

$$B''_{t+i} = R_{pe}\left(B^m_{t+i}\right) \tag{6}$$

where $B^n_{t+i}$ and $B^m_{t+i}$ are the outputs of the modules in the previous layer of the encoder and decoder, $R_{pr}$ is the Patch Reduction operation, and $R_{pe}$ is the Patch Expansion operation.

After fusing the output features of the corresponding layer in the decoder and the output from the corresponding layer in the encoder, different degrees of upsampling are performed respectively, and finally, the upsampled features are spliced and fed into the linear layer to be projected to the d-dimension. The expression for this is

$$F_{i+i} = H_{concat}\left(B^1_{t+i} + B^2_{t+i} \uparrow + B^3_{t+i} \uparrow\right) \tag{7}$$

where $B^1_{t+i}$ and $B^2_{t+i}$ are the outputs of each level of the decoder, respectively, $B^3_{t+i}$ is the output of the bottleneck features, and $F_{t+i} \in R^{W' \times H' \times 3d}$ is the spliced features, and the feature-extracted video frames $I_{t+i} \in R^{W' \times H' \times d}$ are obtained after linear layers, which are then used for spatio-temporal modeling.

With this encoder–decoder network, the output features contain multi-scale information, which is favorable for dealing with the non-uniform blurring of goods.

### 2.2. Alignment and Fusion of Video Frames

2.2.1. Time-Interaction Attention Mechanism

In video deblurring, finding sharp, complementary information in neighboring frames can increase the deblurring performance. To better solve the alignment problem of commodity video frames, this paper uses a Transformer to establish correspondence between reference frames and neighboring frames; furthermore, it proposes a TI-MSA mechanism to pay attention to these frames for potential frame reconstruction. The basic idea is to perform temporal attention across multiple frames at the same location to collect the sharp information of neighboring frames.

As shown in Figure 4, in the temporal interactive attention mechanism of this paper, let the output video frame in the $i - 1$th layer be $\tilde{F}_{i-1,t}$, the $t$-th input video frame in the $i$-th layer be $F_{i,t}$, and the $t + 1$-th input video frame be $F_{i,t+1}$ ($i$ is the $i$-th layer of the temporal interactive attention mechanism). $QK$ and $V$ represent the query element, the key element, and the value of the input video frame, respectively. The purpose of the temporal interaction attention module is to perform remote spatio-temporal modeling and capture the local similarity of neighboring frames. Therefore, the module generates key elements from similar and clearer scene blocks in the spatio-temporal neighborhood.

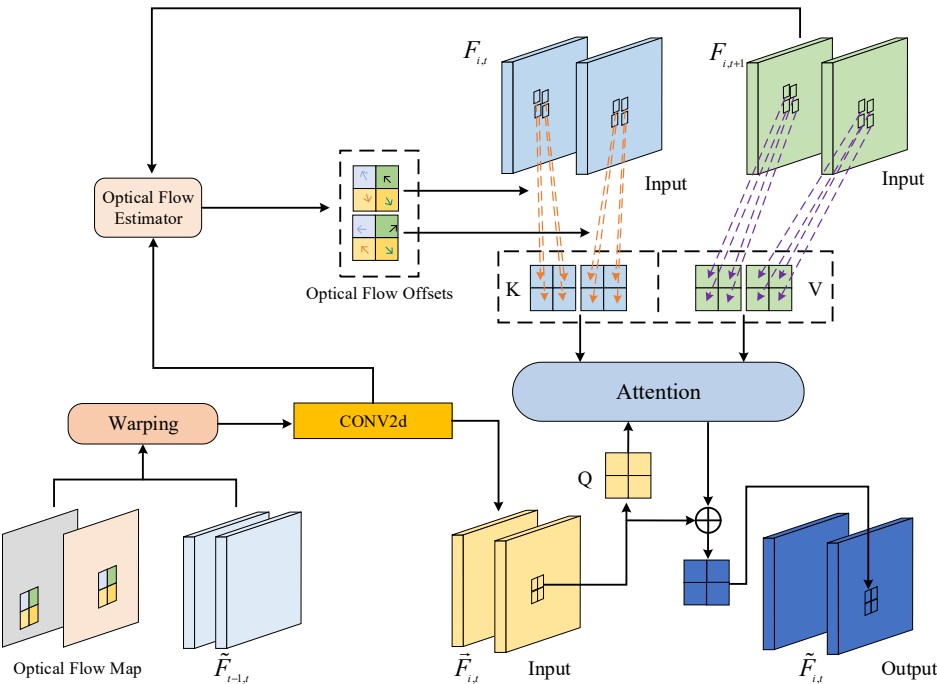

**Figure 4.** Schematic diagram of time interactive attention.

In the process of extracting key elements, in this paper, the output of the previous layer combined with the optical flow schematic is firstly subjected to a warp operation, and afterward, a $3 \times 3$ convolution operation; the formula is expressed as

$$\overrightarrow{F}_{i,t} = Conv\left( W\left( \widetilde{F}_{i-1,t}, O_{i-1} \right) \right) \tag{8}$$

In Equation (8), $O_{i-1}$ denotes the optical flow schematic of the aggregated frame and the reference frame and W denotes the warp operation. This obtains the upper layer aligned aggregation feature map $\overrightarrow{F}_{i,t}$.

After obtaining the aggregated frame features, in this paper, the reference frame and the aggregated frame are subjected to optical flow estimation to predict their optical flow offsets, which can be expressed as

$$o_i = f_w\left( \overrightarrow{F}_{i,t}, F_{i,t+1} \right) \tag{9}$$

where $f_w$ is the mapping function of the optical flow estimation network. In this paper, we choose to use PWC-Net [24] to extract the optical flow information so that the optical flow offset obtained contains more rich video frame sequence motion information.

Subsequently, the inter-frame relative motion vectors from the optical flow offsets can index the corresponding key elements from the neighboring frames. They are denoted as

$$Q_{j,k} = S\left( \overrightarrow{F}_{i,t} P_Q \right) \tag{10}$$

$$K_{j,k} = S\left( F_{i,t} P_K, o_{j,k} \right) \tag{11}$$

$$V_{j,k} = S\left( F_{i,t+1} P_V, o_{j,k} \right) \tag{12}$$

where $S$ is the sampling operation and $P_Q, P_K, P_V$ is the projection matrix. Like in Equation (1), the input video frame is linearly projected through the projection matrix to generate the projection features and is then sampled according to the offset $o_{j,k}$ at the position $(j, k)$ so that the query element corresponds to the similar key element and a clear scene is obtained.

Finally, the query element and key element can be obtained after going through the attention module

$$Attention(Q_{i,t}, K_{i,t}, V_{i,t}) = SoftMax\left(\frac{QK^T}{\sqrt{d}}\right)V \tag{13}$$

Then, the outputs that have gone through the attention mechanism are connected through the residual structure to obtain the following result:

$$X_{i,t} = Attention(LN(Q_{i,t}), LN(K_{i,t}), LN(V_{i,t})) + Q_{i,t} \tag{14}$$

where $LN$ is the normalized layer. Finally, the module adds two fully connected layers and a GELU activation function between the layers to realize the channel interaction, denoted as

$$\widetilde{F}_{i,t} = FFN(X_{i,t}) + X_{i,t} \tag{15}$$

Through the above steps, the aggregated features $\widetilde{F}_{i,t}$ that have been aligned with a multi-frame by the temporal interaction attention module are finally obtained.

Up to this point, this paper utilizes the features of the temporal interaction attention module to establish the communication between adjacent frames so as to achieve better cross-frame information interaction.

### 2.2.2. Feature Recurrent Fusion Mechanism

In order to be able to establish the global modeling relationship of video frames, this paper proposes a feature recurrent fusion mechanism.

The temporal interaction attention mechanism is constructed with connections between neighboring frames, and the feature loop fusion mechanism proposed in this paper makes it possible to model the video frames further away from each other. As shown in Figure 5, in a video clip with $N$ video frames, in this paper, each temporal interaction attention mechanism output is processed and connected to the input of the next one, and the updated video frames are connected to the next attention mechanism, which can be represented as

$$\widetilde{F}_t^{n+1} = TI - MSA\left(\widetilde{F}_t^n, F_t, F_{t+1}\right), n = 1, 2, \ldots, i \tag{16}$$

where $n$ denotes the $n$-th time interactive attention mechanism, $\widetilde{F}_t^n$ is its output, and $F_t$ and $F_{t+1}$ are neighboring reference frames. The update is performed in this way, conveying the information of past frames and establishing the correlation of global video frames, allowing for the addition of valid information.

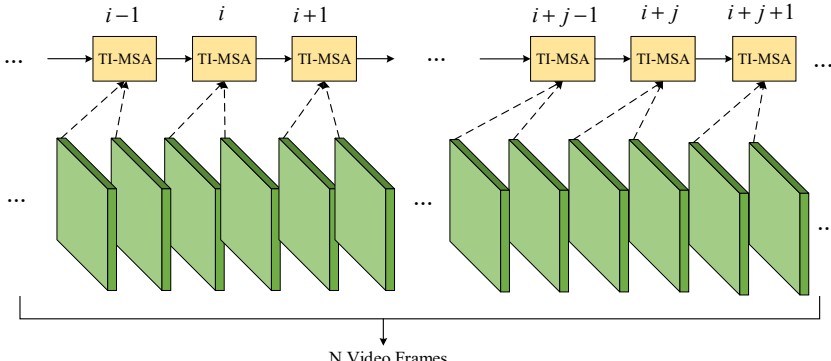

**Figure 5.** Feature recurrent fusion mechanism.

### 2.3. Video Frame Reconstruction

After obtaining the multi-frame aggregated features that have gone through the above temporal interactive attention mechanism and feature loop fusion approach, they are

reconstructed into potentially sharp frames using a global residual learning strategy. In the video frame reconstruction module, the expression is

$$L_t = G_r\left(\widetilde{F}_t^N\right) + B_t \tag{17}$$

where $G_r(\cdot)$ is the reconstruction module, consisting of multiple RSTBs ($N = 20$), two PixelShuffle layers for upsampling, and a linear projection layer that projects the upsampled features into the RGB map.

### 2.4. Loss Functions

In this paper, Charbonnier Loss from the literature [25] is utilized as an optimization objective to obtain friendly visual recovery results. It is represented as

$$L_c = \sqrt{\|I_t - L_t\|^2 + \varepsilon^2} \tag{18}$$

In the formula, $L_t$ is the reconstructed video frame and $I_t$ is the corresponding real video frame. In order to stabilize the values, the $\varepsilon$ of training is set to 0.001 in this paper.

### 3. Experiment Result and Analysis

#### 3.1. Experimental Platform and Dataset

In this experiment, the system platform used is Windows 10, the GPU model is an NVIDIA GeForce RTX3060 with an i5-12400F processor and 16GB RAM, and the software environment is Python3.7.4 and Pytorch2.3.1.

#### 3.2. Dataset Introduction

3.2.1. Fuzzy Commodity Dataset

Currently, the datasets used in motion deblurring algorithms can be roughly divided into three categories. They are (1) synthesized by algorithms, (2) obtained by post-processing of camera acquisition, and (3) obtained by real acquisition. In the field of image motion deblurring, publicly available datasets of motion-blurred commodity images do not yet exist. The training of deep learning networks usually requires a large number of corresponding clear-blurred image datasets. However, due to external factors, it is difficult to acquire a pair of clear and blurred merchandise images under the same location, illumination, and other conditions. Therefore, this paper adopts the method in the literature [26] to synthesize motion blur datasets by simulating the generation of motion trajectories. The process can be modeled as

$$B(x, y) = I(x, y) * K \tag{19}$$

$$K = S(x) \tag{20}$$

where $B(x, y)$ is a blurred image, $I(x, y)$ is a clear image, and $K$ represents the fuzzy kernel. $x$ represents a random trajectory vector generated by a Markov random process. $S(x)$ is the computation of $x$ using a sub-pixel interpolation algorithm.

In Equation (20), the blur kernel is obtained in this paper for different motion modes. After that, in Equation (19), the obtained fuzzy kernel is subjected to convolution operation with a clear image, i.e., the fuzzy kernel is slid over the image, weighted, and summed with the pixel values of the image at each position to obtain the blurred image.

Due to the random nature of the generated motion trajectories, the synthesized images contain blurred scenes under various conditions, such as camera shake and uniform linear motion. Therefore, it is reasonable to use synthetic datasets in this way to train deep-learning network models.

Here, since this paper uses a video deblurring dataset, a sequence of clear video frames is first obtained, and then an algorithm is used on each video frame, which, in turn, generates a blurred video. The dataset was produced with a total of 11 blurred and clear video clips of 11 commodities, with a total of 11 blurred videos and corresponding clear

videos (where each video clip contains 50 video frames). The experiments were conducted by dividing the video frames into a training set and a test set according to a ratio of 4:1. Among them, 440 pairs of video frames were used as the training set and 110 pairs of video frames were used as the test set. It should be noted that the training and test sets are non-repetitive, i.e., there are no identical video frames between them, as shown in Figure 6 for the partial Fuzzy Commodity Dataset (FCD).

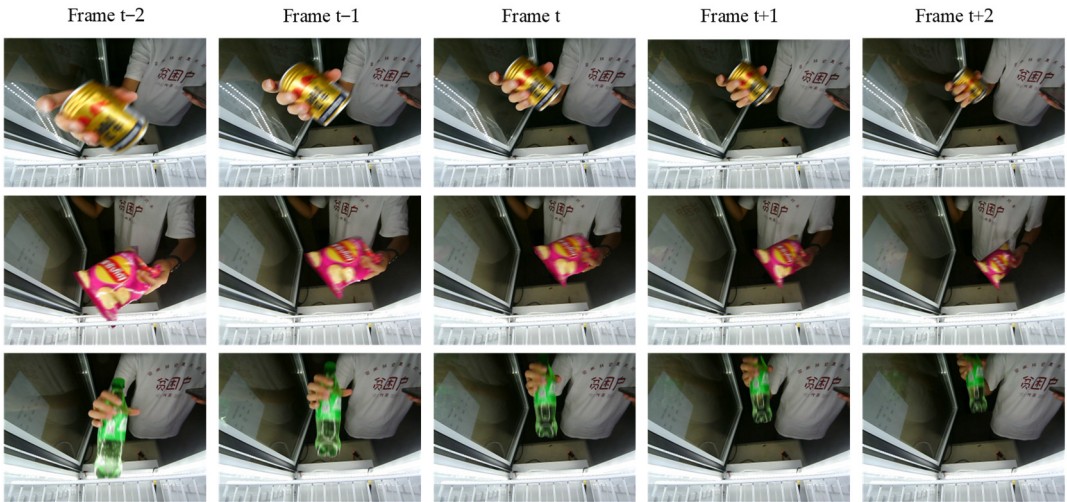

**Figure 6.** Blurred video frames of the commodity pickup process.

### 3.2.2. Public Dataset

This article conducts experiments on a Deep Video Deblurring (DVD) video deblurring dataset [9]. The data configuration is as follows. The DVD dataset has 71 blurry and correspondingly clear videos, of which 61 are used for training and 10 for testing (each test video evaluating 10 frames).

### *3.3. Controlled Experiment*

### 3.3.1. Evaluation Index

In this experiment, this paper uses the Peak Signal-to-Noise Ratio (PSNR) and Structural Similarity (SSIM) [27] metrics to evaluate the quality of an image or video. They are typically used to compare the degree of difference between two images or videos. The PSNR is a common measure of image or video quality that quantifies image quality by comparing the difference between the original image (or video) and the compressed or processed image (or video).

The calculation formula for the PSNR is as follows:

$$PSNR = 10 * lg\left(\frac{2^L - 1}{MSE}\right) \tag{21}$$

where $L$ denotes the maximum gray level; if the maximum gray value is 255, then $L$ is 8, and $MSE$ denotes the mean square error. A larger $PSNR$ value means a better deblurring effect.

SSIM is an image quality assessment metric that combines luminance, contrast, and structural information. SSIM measures the degree of similarity between the original and reconstructed images by comparing their luminance, contrast, and SSIM.

$$SSIM(X,\ Y) = \frac{(2\mu_X\mu_Y + C_1)(2\sigma_{XY} + C_2)}{(\mu_X{}^2 + \mu_Y{}^2 + C_1)(\sigma_X{}^2 + \sigma_Y{}^2 + C_2)} \tag{22}$$

$X$ and $Y$ denote the two input images, respectively, $\mu_X, \mu_Y$ denotes the mean value of the two images, $\sigma_X,\ \sigma_Y$ denote the standard deviation of the two images, respectively, $\sigma_{XY}$ denotes the covariance of $X$ and $Y$, and $C_1,\ C_2$ is a constant. Usually, taking $C_1(K_1 * L)^2$

and $C_2(K_2 * L)^2$, in general, $K_1 = 0.01$, $K_2 = 0.03$, and $L = 255$ (dynamic range of pixel values). $SSIM$ takes values ranging from 0 to 1. The more similar the images are, the closer the result is to 1.

These two metrics are widely used in the fields of image or video compression, enhancement, and restoration.

### 3.3.2. The Details of the Training

During the training process, the algorithm of this paper is implemented in the PyTorch framework. In this paper, a pre-trained optical flow network is used as the optical flow estimation, and all the modules in the algorithm are trained for 300 epochs using Adam [28] as the optimizer ($\beta_1 = 0.9$, $\beta_2 = 0.99$). The initial learning rate of the video deblurring network and the optical flow estimation network. PWC-Net in this paper are $2 \times 10^{-4}$ and $2.5 \times 10^{-5}$, respectively, and the learning rate decreases to half of the original one for every 100 iterations during the training process. The input is a short video clip in RGB format consisting of three consecutive blurred frames, and the algorithm first performs random cropping (patch size of $256 \times 256$) and flipping (horizontally and vertically) to augment the data. The models were performed on a GPU model, an NVIDIA GeForce RTX3060, with 12 G of RAM.

### 3.3.3. Comparative Study of Different Algorithms

In this paper, the proposed model, named the Commodity Video Deblurring Transformer (CVDT), is compared with several advanced deep learning algorithm models on DVD datasets in recent years, including EDVR [14], TSP [12], ARVo [15], and the Video Restoration Transformer (VRT) [22]. It is evaluated using a publicly available source code using PSNR and SSIM as evaluation metrics.

In Table 1, it can be concluded that compared with other CNN-based algorithms, this paper's algorithm is far ahead of other CNN-based algorithms in deblurring the DVD dataset because the Transformer achieves a good relationship between spatial and temporal modeling. Compared with the recent Transformer-based deblurring algorithm VRT, the PSNR value of this paper is 0.23 dB higher and the SSIM value is 0.011 higher, which is due to the fact that this paper's algorithm pays more attention to the global video frame relationship and constructs the spatio-temporal modeling relationship in a better way.

**Table 1.** Comparison of the deblurring ability of different algorithms on the DVD dataset.

| Algorithms | PSNR | SSIM |
| --- | --- | --- |
| EDVR | 31.82 | 0.916 |
| STFAN | 31.15 | 0.905 |
| ARVo | 32.80 | 0.935 |
| VRT | 33.13 | 0.936 |
| **CVDT** | **33.36** | **0.947** |

Note: Bold is the best result.

In addition to this, the algorithm of this paper is compared with other algorithms on our homemade dataset, and Table 2 shows the experimental results.

As shown in Table 2, when the deblurring effect of the technique used in this research is compared to other algorithms in the FCD dataset, it is evident that both the PSNR and SSIM values are optimal. The algorithm module proposed in this paper's algorithm is more adept at dealing with some complex motion blurring situations generated by commodities with different degrees of motion and thus achieves optimal results.

**Table 2.** Comparison of the deblurring ability of different algorithms on the FCD dataset.

| Algorithms | PSNR | SSIM |
|:---:|:---:|:---:|
| EDVR | 27.62 | 0.796 |
| STFAN | 26.97 | 0.784 |
| ARVo | 27.73 | 0.853 |
| VRT | 28.16 | 0.811 |
| **CVDT** | **28.97** | **0.833** |

Note: Bold is the best result.

### 3.3.4. Dataset Visualization Comparison

In the results of the DVD dataset shown in Figure 7, it can be seen that other algorithms recover the test data in general; they either produce excessively smooth images at the expense of fine structural content and texture details or they lead to redundant speckled textures and color artifacts. In contrast, the algorithm in this paper recovers more structural content and texture details, and the deblurring ability effect is closest to the real situation. In addition, we further estimate the optical flow between two neighboring prediction frames on the DVD dataset, and it can be seen that with the temporal interactive attention and feature loop fusion mechanism of this paper's algorithm, the optical flow results in this paper are also closer to the real situation, indicating better motion consistency.

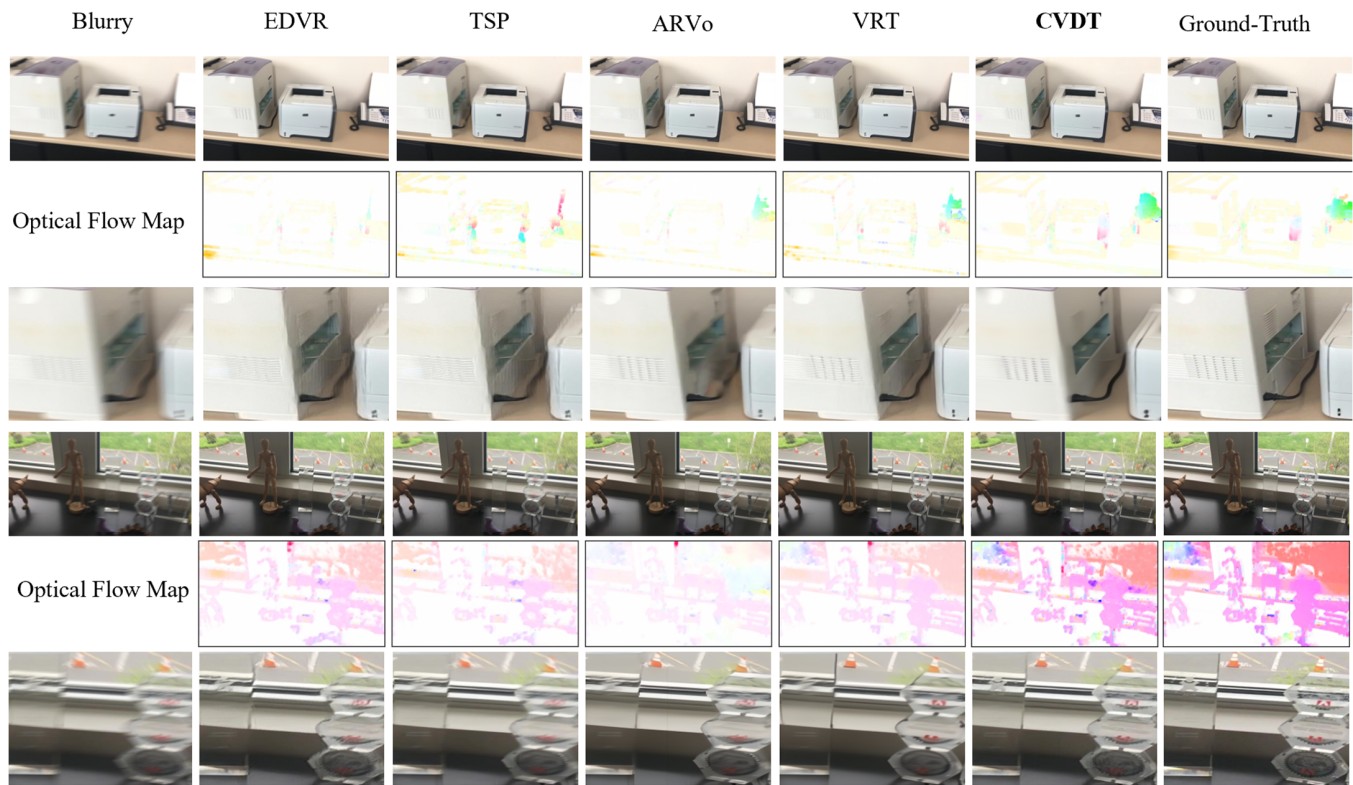

**Figure 7.** Comparison of the deblurring effects and an Optical Flow Map using different algorithms on DVD datasets. Note: Bold is the best result.

In the FCD dataset in Figure 8, it can be observed that when dealing with fuzzy merchandise, the comparison network has some details lost in the recovery process. In addition, the edge processing of the merchandise target is not clear enough and too smooth, which is not favorable for the detection of the merchandise. In addition, due to the problem of the low resolution of the small commodities themselves, several algorithms show different degrees of over-smoothing on the degree of recovery of the commodity target when recov-

ering motion-blurred video frames. The method in this paper alleviates this phenomenon to a certain extent and achieves the best results from subjective visual perception.

Blurry EDVR TSP ARVo VRT **CVDT** Ground-Truth

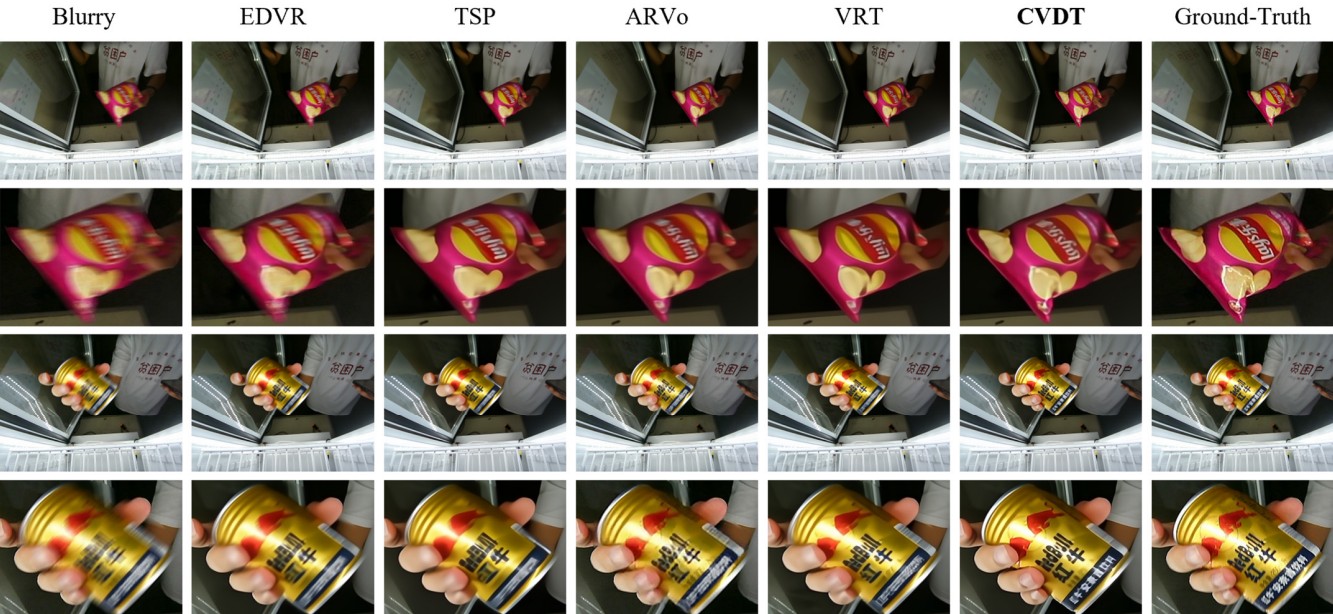

**Figure 8.** Comparison of the deblurring effects using different algorithms on the FCD dataset. Note: Bold is the best result.

### 3.4. Ablation Study

3.4.1. Multi-Scale Transformer Structure Experiment

The multi-scale Transformer structure in this paper's algorithm contains two upsamplings and two downsamplings, and the number of RSTBs passing through each upsampling and downsampling is three. Relevant ablation experiments have been carried out to validate the effectiveness of the proposed new module and idea. According to the different numbers of RSTBs in the multi-scale Transformer, this paper categorizes them into four subsets, as shown in Table 3 Here, the experiments are carried out on the FCD dataset, and how the number of RSTBs in the multi-scale Transformer should be allocated is analyzed by comparing the PSNR and SSIM values. From the results in Table 3, it can be seen that the algorithm deblurring ability is optimal when the number of RSTBs is three.

**Table 3.** Comparison of deblurring ability among different RSTB quantities.

| RSTB | PSNR | SSIM |
|------|------|------|
| 1 | 28.44 | 0.814 |
| 2 | 28.53 | 0.817 |
| **3** | **28.97** | **0.833** |
| 4 | 28.72 | 0.821 |

Note: Bold is the best result.

Secondly, the algorithm in this paper, with the number of RSTBs being three, can divide the multi-scale Transformer into four subsets as well, depending on the number of sampling layers. As shown in Table 4, comparing the PSNR and SSIM values, it can be seen that the two upsampling and downsampling Transformer structures are more conducive to feature extraction for non-uniform fuzzy commodity video frames.

**Table 4.** Comparison of deblurring ability among different sampling times.

| Times | PSNR | SSIM |
|---|---|---|
| 0 | 27.07 | 0.788 |
| 1 | 28.22 | 0.815 |
| **2** | **28.97** | **0.833** |
| 3 | 28.70 | 0.820 |

Note: Bold is the best result.

### 3.4.2. Performance Experiment of the TI-MSA Module

To validate the deblurring ability of the temporal interaction attention mechanism, the proposed module is compared with other Transformer-based attention mechanisms. Here, the comparison is performed with the global self-attention mechanism (MSA) model, the local window self-attention mechanism (W-MSA) model, the TMSA model [22], the TI-MSA model in this paper, and the TI-MSA+, which has a feature cyclic fusion mechanism. The experiments are conducted on the FCD dataset. Table 5 shows the experimental results.

**Table 5.** Comparison of deblurring ability among different attention mechanisms.

| Mechanisms | PSNR | SSIM |
|---|---|---|
| MSA | 27.08 | 0.789 |
| W-MSA | 27.62 | 0.803 |
| TMSA | 28.16 | 0.811 |
| TI-MSA | 28.82 | 0.827 |
| **TI-MSA+** | **28.97** | **0.833** |

Note: Bold is the best result.

According to Table 5, for the MSA, the PSNR value is reduced by 1.89 dB compared to this paper's algorithm. This is mainly due to the fact that the MSA deals with key elements that are too redundant and require abundant computational and memory resources and at the same time leads to gradient blurring of the input features, which results in non-convergence problems, while in the W-MSA, it is reduced by 1.35 dB compared to the algorithm in this paper. The acceptance range is limited because it computes self-attention within a window of a specific location and cannot be computed across frames. The TMSA is reduced by 0.66 dB compared to the TI-MSA, which indicates that the temporal interactive attention mechanism in this paper can better align the features to better capture the information of each frame in the merchandise video sequence. With the inclusion of the feature loop fusion mechanism, which is able to better utilize the relationship between the merchandise video frames, the deblurring effect of this paper's algorithm is better compared to the previous one, with a higher PSNR value of 0.15 dB.

### 3.5. Motion Blur Commodity Detection

In order to evaluate the practical application capability of this paper's algorithm, the YOLOv5 target detection algorithm was used to detect the test images, in which the images include clear images, motion blur images, and blur recovery images with different algorithms. In addition, the experiment is only used to explore the effect of this paper's algorithm on target detection accuracy. Since the differences in the results are not significant on different target detection methods, the application of other classical target detection algorithms on this test image is not further investigated in this paper.

Figure 9 shows a comparison graph of the results of the YOLOv5 target detection algorithm for merchandise detection on different restoration algorithm images. According to the detection result graph, it can be observed that this paper's algorithm achieves optimal results for merchandise detection results. Through the deblurring process, it solves the leakage and incomplete detection problems while improving the accuracy of detection.

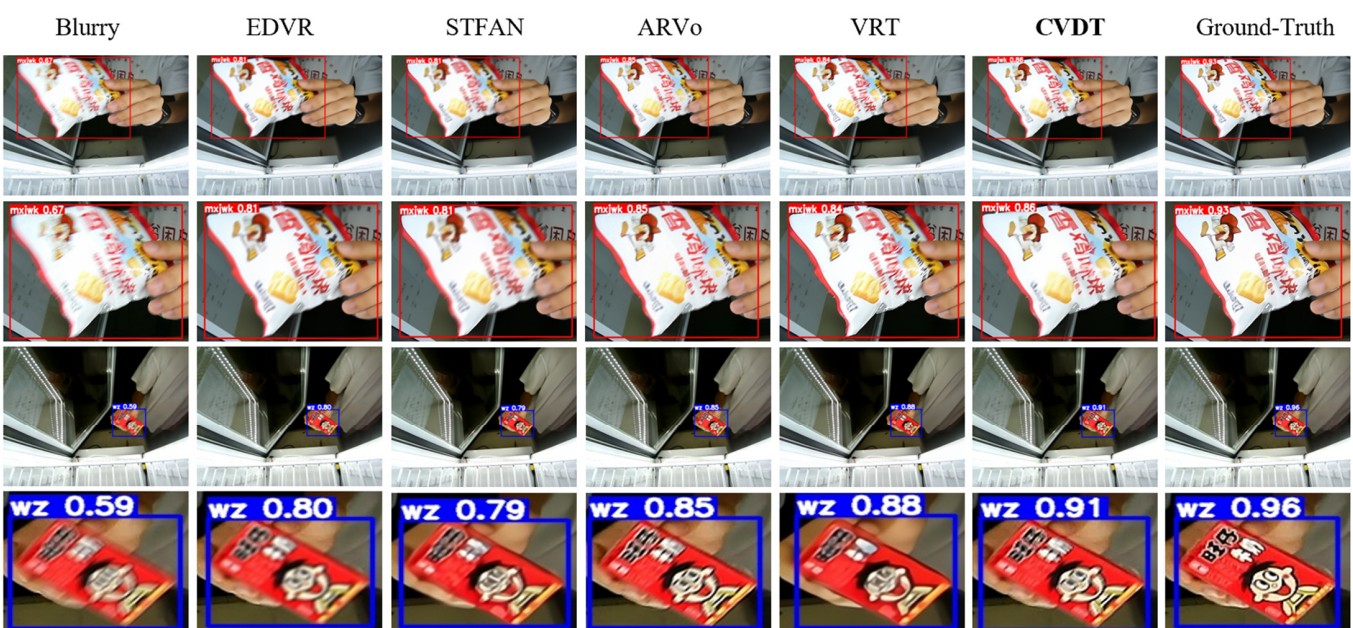

**Figure 9.** Comparison of commodity detection results on restored images using different algorithms. Note: Bold is the best result.

## 4. Discussion and Conclusions

In order to effectively eliminate the fuzzy features of goods in visual cabinets, a new Transformer-based video deblurring network is proposed in this paper. In this network, deep feature extraction with a multi-scale Transformer structure is first adopted to handle the complex non-uniform blur features of commodities. Then, a temporal interactive attention mechanism is designed to focus on different important information by assigning different weights in adjacent frames of the commodities, which, in turn, aggregates effective features from the unaligned fuzzy commodity video frames. Finally, this paper proposes a feature recurrent fusion mechanism to establish a global link between merchandise video frames to complement the useful information in reconstructed merchandise video frames.

Experimentally, this paper is evaluated on a DVD dataset and a homemade dataset (FCD), verifying the effectiveness of the algorithm in this paper. In addition, the ablation study demonstrates the design of the modules of the algorithm. Finally, this paper also performs detection on recovered fuzzy images of merchandise, and the results indicate that it can improve the accuracy of merchandise detection in practical applications.

Since the homemade dataset used in this paper still has some limitations, subsequent research will focus on the commodity holding process under complex conditions, further improving the dataset in the experiments, optimizing the computational complexity of the model, and achieving an accurate and highly efficient detection of small commodities.

**Author Contributions:** Conceptualization, S.H., Q.L. and K.X.; methodology, S.H. and Q.L.; software, Q.L.; writing—original draft preparation, S.H.; writing—review and editing, S.H.; visualization, Q.L. and J.H.; investigation, Z.H., W.Z. and C.W.; project administration, K.X.; funding acquisition, J.H. and W.Z. All authors have read and agreed to the published version of the manuscript.

**Funding:** This research was supported by the National Natural Science Foundation of China (Grant No. 62272485) and the National Natural Science Foundation of China (Grant No. 62373372). In cooperation with them, the work in this paper can be successfully established and carried out to proceed. This work is also part of the Undergraduate Training Programs for Innovation and Entrepreneurship at Yangtze University under Grant Yz2023056, which focuses on the study of consumer behavior in dynamic visual cabinets. And, the research work in this paper related to commodity inspection in dynamic visual cabinets is carried out under the support of the National Innovation and Entrepreneurship Training Program for College Students under Grant No. 202310489005.

**Institutional Review Board Statement:** Not applicable.

**Informed Consent Statement:** Informed consent was obtained from all subjects involved in the study.

**Data Availability Statement:** Data are unavailable due to privacy.

**Conflicts of Interest:** Author Zhengfang He was employed by the company Zhejiang Uniview Technologies Co., Ltd. The remaining authors declare that the research was conducted in the absence of any commercial or financial relationships that could be construed as a potential conflict of interest.

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
