# Peer review of "Improved Transformer-Based Deblurring of Commodity Videos in Dynamic Visual Cabinets"

_electronics, doi:10.3390/electronics13081440_

Round 1

Reviewer 1 Report

Comments and Suggestions for Authors

Attached

Comments on the Quality of English Language

Moderate proof-reading is essential

Author Response

Please see the attached file: Response to Reviewers 1.docx.

Reviewer 2 Report

Comments and Suggestions for Authors

In this paper, Transformer-based video deblurring network are presented and concludes that this network can improve commodities detection accuracy in Dynamic Visual Cabinets (DVC) according to results obtained on two datasets, one public and one homemade. The article provides a clear and adequate methodology and experimentation. However, the definition of Dynamic Visual Cabinet is not clear, and its operation is difficult to understand. Figures and tables lack descriptive captions. Several abbreviations of terms are presented, but some of them are not used in the rest of the document. In the Evaluation index section, two metrics PSNR and SSIM are presented, however, the equations and possible values that can be obtained are not shown, which makes it difficult to understand the results presented. The equations presented in the paper are not adequately described, mainly 19 and 20. It is mentioned that the goal of DVC is the detection of the commodities in motion, in the subsection Motion blur commodity detection a comparison of the proposed network against the other blur algorithms presented in the paper would be expected. A revision of the English used in the document is required.

Author Response

Please see the attached file: Response to Reviewers 2.docx.

Round 2

Reviewer 2 Report

Comments and Suggestions for Authors

In this paper Transformer-based video deblurring network are presented and it concludes that this network can improve commodities detection accuracy in Dynamic Visual Cabinets (DVC) according to results obtained on two datasets, one public and one homemade. The article provides a clear and adequate methodology and experimentation.

I appreciate your responses, which I find satisfactory. Therefore, I consider your research and contribution suitable for MDPI. However, I consider that the caption in Figure 1 should be expanded following the guidelines of the MDPI template that it mentions: “If there are multiple panels, they should be listed as: (a) Description of what is contained in the first panel; (b) Description of what is contained in the second panel. Figures should be placed in the main text near to the first time they are cited”

Author Response

Please see the attached file:Response to Reviewers 2.docx
